# VOLD: Reasoning Transfer from LLMs to Vision-Language Models via On-Policy Distillation

## Abstract

Training vision-language models (VLMs) for complex reasoning remains a challenging task, i.a. due to the scarcity of high-quality image-text reasoning data. Conversely, text-based reasoning resources are abundant and scalable, but it is still an open question how to leveraging them for VLM reasoning. To address this problem, we propose VOLD, a framework to transfer reasoning capabilities from text-only teacher models to VLM student models. To this end, VOLD combines reinforcement learning via Group Relative Policy Optimization (GRPO) with on-policy distillation, which allows the student reasoning traces to be guided by the teacher model, resulting in a significant gain over using GRPO alone. We further show that a cold-start alignment is essential for an effective transfer during the online training phase in this scenario and that without sufficient distributional alignment between teacher and student, on-policy distillation fails to provide meaningful guidance. We evaluate VOLD across diverse benchmarks including MMMU-Pro, MathVision, MathVista, and LogicVista, showing that VOLD outperforms the baseline model significantly and improves over the state of the art by a margin. Our ablation shows the importance of a cold-start alignment via SFT for on-policy distillation with a text-only teacher.

## 1 Introduction

The remarkable success of text-based reasoning models can be attributed in part to their ability to leverage vast quantities of text-based reasoning traces for bootstrapping. As a result, there is growing interest within the research community in exploring how such reasoning capabilities might be extended to other modalities, notably vision. However, the acquisition of vision-language data to train reasoning models presents significant challenges. While numerous image-text datasets exist, few provide the complexity required for vision-based reasoning training, as most samples are limited to basic perception tasks (e.g., identifying objects on a sofa) rather than demanding multi-step reasoning. In contrast, the collection of text-based reasoning data for domains such as mathematics or programming has proven both feasible and scalable for training models via reinforcement learning (RL), as demonstrated by recent advances in models like DeepSeek-R1 and QwQ. This scalability advantage stems from the ability to automatically generate and verify text-based reasoning traces, whereas visual reasoning data curation remains labor-intensive and difficult to automate.

Existing approaches address VLM reasoning training through several strategies. One line of work creates synthetic visual reasoning traces by augmenting text-based reasoning with visual descriptions (Vision-R1 (Huang et al., 2025), OpenVLThinker (Deng et al., 2025), R1-VL (Zhang et al., 2025)). Another approach involves collecting challenging samples from existing benchmarks for training, as demonstrated by VLAA-Thinker and VLM-R1, though this strategy requires careful consideration of evaluation protocols to ensure fair comparison across different test sets. Alternatively, some methods explore training on text-only data for reasoning transfer, such as X-Reasoner (Liu et al., 2025a), which is the direction our approach follows. However, these text-based transfer methods do not fully leverage the teacher models used to generate the reasoning traces, missing opportunities for ongoing guidance during training.

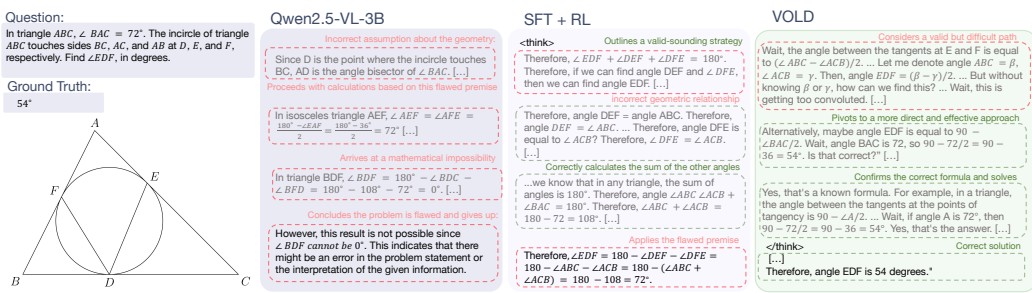

Figure 1: **Visual Reasoning Examples**. *(left)* The base model fails the task due to a flawed geometric assumption. *(center)* The base model trained with SFT+RL only-on text outlines a valid plan but uses an incorrect formula, leading to a wrong answer. *(right)* The model trained with SFT+RL and guided by on-policy distillation from a teacher LLM successfully navigates the problem. It demonstrates flexible reasoning by considering and then discarding a difficult approach in favor of a more direct and correct one.

Meanwhile, advances in text-to-text reasoning transfer have demonstrated the effectiveness of combining reinforcement learning with teacher distillation. KDRL Xu et al. (2025a) and Qwen3 Yang et al. (2025a) show that on-policy knowledge distillation significantly improves RL sample efficiency by providing additional teacher-based guidance during training. We build on this insight to develop a unified framework for text-to-vision reasoning transfer that aligns text-only teachers with VLM students through coordinated RL and distillation objectives.

To this end, we propose VOLD, a framework that transfers reasoning capabilities from text-only teacher models to vision-language student models using purely text-based training data as shown in Figure 1. To enable the effective reasoning transfer VOLD combines GRPO reinforcement learning with on-policy knowledge distillation, enabling VLMs to develop reasoning capabilities for visual tasks without requiring vision-based reasoning data during training. It shows that a successful text-to-vision reasoning transfer requires initial policy alignment between teacher and student models. Our framework therefore starts with an SFT cold-start phase, training the VLM on reasoning traces generated by the text-only teacher to establish distributional alignment. This enables the VLM to benefit from rich text-based reasoning data while avoiding dependence on limited visual reasoning resources. Figure 2 illustrates our two-stage training pipeline. VOLD uses Qwen2.5-VL-3B as the student VLM and Qwen3-8B as the text-only teacher. Stage 1 involves generating reasoning traces from the teacher on mathematical reasoning prompts and performing SFT to align the student with the teacher's reasoning distribution. Stage 2 implements our unified objective combining GRPO and on-policy distillation on text-only reasoning tasks.

The final model is evaluated in zero-shot mode, without any further fine-tuning on image-text data, on multiple challenging visual reasoning datasets—MMMU-Pro (Yue et al., 2025), MMStar, Math-Vision (Wang et al., 2024a), MathVista (Lu et al., 2024), MathVerse(Zhang et al., 2024), Dyna-Math(Zou et al., 2025), WeMath(Qiao et al., 2024) and LogicVista(Xiao et al., 2024)—demonstrating successful reasoning transfer from text-only training to visual tasks. Our evaluation shows that the cold-start alignment phase is essential for effective text-to-vision reasoning transfer, enabling the student to benefit from teacher guidance during unified training. Importantly, VOLD is orthogonal and complementary to most other current approaches in RL, allowing the proposed unified framework to be seamlessly integrated with any improved RL method beyond vanilla GRPO.

The main contributions of this work are summarized as follows:

1) We propose **VOLD**, a framework for transferring reasoning capabilities from text-only teachers to vision-language students through combined reinforcement learning and on-policy knowledge distillation. 2) We show that text-to-vision policy alignment via SFT is a critical prerequisite for effective teacher-student reasoning transfer and provide a theoretical motivation as well as an empirical validation for this requirement. 3) We demonstrate that unified RL and distillation objectives significantly outperform standalone GRPO training, achieving state-of-the-art performance despite using only text-based training data.

## 2 RELATED WORK

**Reasoning in LLMs** The advancement of reasoning capabilities in language models represents a critical frontier for solving complex, multi-step problems requiring sophisticated logical thinking. Early symbolic and rule-based approaches (Newell & Simon, 1976; Nilsson, 1980) suffered from brittleness and limited scalability. A transformative shift occurred with OpenAI's o1 model (Jaech et al., 2024), which pioneered explicit thinking traces before final answers, demonstrating that RL could effectively train human-like deliberative reasoning in LLMs. However, the proprietary methodology limited broader research progress. The landscape changed when DeepSeek released DeepSeek-R1 (Guo et al., 2025), providing the first open-source implementation with a reproducible recipe. DeepSeek-R1 established the foundational paradigm combining SFT on reasoning traces followed by RL using Group Relative Policy Optimization (GRPO) (Shao et al., 2024) on verifiable mathematics and coding problems. This sparked numerous follow-up works including Dr.GRPO (Liu et al., 2025b), DAPO (Yu et al., 2025), DCPO (Yang et al., 2025b), and others, each contributing specialized techniques for improving sample efficiency and training stability in reasoning-focused RL.

**Knowledge Distillation for LLM** Knowledge distillation for LLMs encompasses two primary strategies. Off-policy distillation uses teacher-generated data to train students, as in LLaMA3 series (Grattafiori et al., 2024) and DeepSeek-R1-Distill (Guo et al., 2025), applying distillation at token-logit levels (Hinton et al., 2015; Sanh, 2019). On-policy distillation (Agarwal et al., 2024) represents a different approach where students generate trajectories and teachers provide feedback on self-generated sequences, mitigating exposure bias and supporting RL paradigms. Recent work like Qwen3 (Yang et al., 2025a) demonstrates its potential for improving reasoning capabilities. Most recently, KDRL (Xu et al., 2025a) demonstrated this integration for text-to-text reasoning transfer, showing significant improvements over standalone RL approaches. Our work takes the innovative path of using a text-only teacher LLM to train a VLM student, representing the first text-to-vision reasoning transfer through unified on-policy distillation and RL.

**VLM reasoning** While established training recipes have proven effective for text-only reasoning models, extending these methodologies to VLMs remains significantly more challenging. Researchers have pursued several distinct approaches to instill VLMs with reasoning abilities. One direction involves synthetic visual reasoning data generation, where a text-based reasoning model is augmented with visual descriptions. Methods like OpenVLThinker (Deng et al., 2025) distill reasoning from text models using image captions, while R1-OneVision (Yang et al., 2025c) transforms images into structured textual representations. However, these approaches struggle with the modality gap, as textual captions provide limited visual representations compared to direct visual perception. An alternative strategy trains on challenging image-text pairs from existing benchmarks. VLAA-Thinker (Chen et al., 2025) advocates for direct GRPO training on challenging samples, while VLM-R1 (Shen et al., 2025) extends rule-based RL to visual tasks. However, this approach faces limitations in finding and scaling high-quality samples while avoiding test data contamination. The most closely related work is X-REASONER (Liu et al., 2025a), which combines SFT with RL on text-only data, demonstrating that text-based post-training can transfer reasoning to visual tasks. Our work advances beyond these approaches by fully leveraging existing text reasoning models through on-policy distillation, providing a unified framework that combines RL with teacher guidance for more effective text-to-vision reasoning transfer.

## 3 METHOD

### 3.1 TECHNICAL PRELIMINARIES AND NOTATION

**Notation** Let $q$ denote an input prompt or query. A trajectory $\boldsymbol{\tau}$ represents a sequence of tokens $(y_1, y_2, \ldots, y_T)$ of length $T$, where $y_t$ is the token at step $t$ and $\boldsymbol{y}_{<t} = (y_1, \ldots, y_{t-1})$ denotes the prefix up to step $t-1$. The state at step $t$ is defined as the history $h_t = (q, \boldsymbol{y}_{<t})$.

We define two policies: the student policy $\pi_\theta(y_t|h_t)$ represents the probability of generating token $y_t$ given history $h_t$, parameterized by $\theta$, while the teacher policy $\pi_\phi(y_t|h_t)$ is parameterized by $\phi$ to distinguish it as a separate, fixed model. We define $r(\boldsymbol{\tau})$ as the scalar reward for a completed trajectory $\boldsymbol{\tau}$. In our framework, rewards are binary: $r(\boldsymbol{\tau}) \in \{0, 1\}$. The sequence-level entropy,

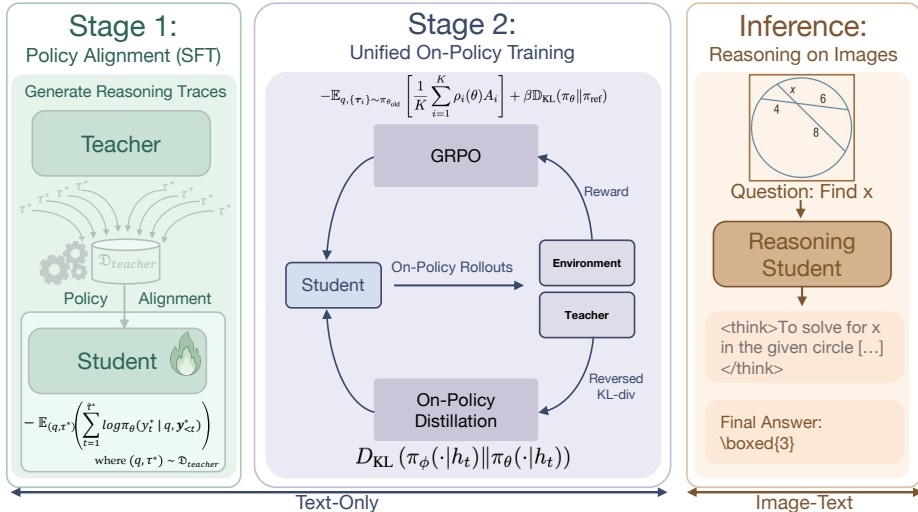

Figure 2: **VOLD training pipeline:** VOLD is a two-stage process to instill reasoning capabilities into a student VLM using a text-only teacher. *(Stage 1)*, the student's policy is aligned with the teacher's via SFT on a corpus of teacher-generated reasoning traces. *(Stage 2)*, the student is trained with a unified on-policy objective that leverages the same rollouts to compute both a sparse reward for RL(GRPO) and a dense distillation loss against the teacher. This combined signal enhances reasoning without requiring any vision-based reasoning data. At Inference, the resulting student model can effectively reason over novel image-text prompts.

averaged per token, is computed as:

$$H(\pi_\theta) = \mathbb{E}_{q, \boldsymbol{\tau} \sim \pi_\theta} \left[ \frac{1}{T} \sum_{t=1}^{T} H(\pi_\theta(\cdot|h_t)) \right] \tag{1}$$

where $H(\pi_\theta(\cdot|h_t)) = -\sum_{y \in \mathcal{V}} \pi_\theta(y|h_t) \log \pi_\theta(y|h_t)$ is the token-level entropy over vocabulary $\mathcal{V}$.

**Group Relative Policy Optimization (GRPO)** GRPO is a value-function-free reinforcement learning algorithm that estimates advantages using intra-group relative comparisons rather than explicit value functions. For each prompt $q$, we sample a group of $K$ trajectories $\{\boldsymbol{\tau}_i\}_{i=1}^{K}$ from the old policy $\pi_{\theta_{old}}$. Let $r_i = r(\boldsymbol{\tau}_i)$ denote the reward for trajectory $i$. The group-relative advantage for trajectory $i$ is computed as: $A_i = \frac{r_i - \bar{r}}{\sigma_r + \delta}$ where $\bar{r} = \frac{1}{K} \sum_{j=1}^{K} r_j$ is the mean reward within the group, $\sigma_r$ is the standard deviation of rewards in the group, and $\delta$ is a small constant for numerical stability.

The importance ratio between the current and old policies is defined as $\rho_i(\theta) = \frac{\pi_\theta(\boldsymbol{\tau}_i|q)}{\pi_{\theta_{old}}(\boldsymbol{\tau}_i|q)}$. The GRPO loss to be minimized is:

$$\mathcal{L}_{\text{GRPO}}(\theta) = -\mathbb{E}_{q, \{\boldsymbol{\tau}_i\} \sim \pi_{\theta_{old}}} \left[ \frac{1}{K} \sum_{i=1}^{K} \min \left( \rho_i(\theta) A_i, \text{clip}(\rho_i(\theta), 1-\epsilon, 1+\epsilon) A_i \right) \right] + \beta \mathbb{D}_{\text{KL}}(\pi_\theta \| \pi_{\text{ref}})$$
$$\tag{2}$$

where $\epsilon$ is the clipping threshold, $\beta$ controls the KL regularization strength, and $\pi_{\text{ref}}$ is the reference model. Following DAPO, we apply asymmetric clipping with $\epsilon_{\text{upper}} = 0.3$ and $\epsilon_{\text{lower}} = 0.2$ to allow greater exploration while maintaining stability. Additional entropy regularization can be applied separately as regularization.

**On-Policy Knowledge Distillation** On-policy knowledge distillation aims to minimize the reverse KL divergence between teacher and student distributions at each step of the student's own generated trajectories. Unlike traditional distillation on fixed datasets, on-policy distillation provides student supervision **on its own sampled trajectories**, where the teacher is queried on the same prefix $h_t$ to provide distributional targets that adapt to the student's evolving policy.

The reverse KL distillation loss is defined as:

$$\mathcal{L}_{\text{RKL}}(\theta) = \mathbb{E}_{q, \boldsymbol{\tau} \sim \pi_\theta} \left[ \sum_{t=1}^{T} D_{\text{KL}} \left( \pi_\phi(\cdot|h_t) \| \pi_\theta(\cdot|h_t) \right) \right] \tag{3}$$

where $D_{\text{KL}}(P\|Q)$ denotes the KL divergence. The expectation $\mathbb{E}_{\boldsymbol{\tau} \sim \pi_\theta}$ makes this "on-policy" since the prefixes $h_t$ come from trajectories sampled from the current student policy $\pi_\theta$. Computing the full-vocabulary KL divergence is computationally expensive. In practice, it is computed using a Monte-Carlo approximation. In this paper, we use the "k2" (Schulman, 2020) estimator, which leverages the log-probabilities of the single token $y_t$ sampled from the student policy.

The KL divergence computation requires that teacher and student share the same tokenizer and vocabulary for meaningful KL computations—a requirement naturally satisfied in modern model families where VLMs inherit the base LLM tokenizer.

## 3.2 VOLD FRAMEWORK

VOLD features a two-stage post-training pipeline designed to transfer reasoning capabilities from text-only teacher LLMs to student VLMs without requiring vision-based reasoning data. The pipeline consists of two sequential stages: Stage 1 performs supervised fine-tuning (SFT) to align the student's output distribution with the teacher's reasoning patterns, while Stage 2 applies a unified objective combining reinforcement learning and on-policy knowledge distillation to enhance reasoning capabilities. Figure 2 provides an overview of the complete framework.

**Stage 1: SFT for Policy Alignment**   The goal of Stage 1 is to reduce the initial policy divergence between the student VLM and teacher LLM, creating a foundation that enables the student to effectively follow the teacher's reasoning process during the on-policy phase.

We construct a synthetic dataset $\mathcal{D}_{\text{teacher}} = \{(q_j, \boldsymbol{\tau}_j^*)\}_{j=1}^{N}$ where each $q_j$ is a reasoning prompt and $\boldsymbol{\tau}_j^*$ is a reasoning trace sampled from the teacher, i.e., $\boldsymbol{\tau}_j^* \sim \pi_\phi(\cdot|q_j)$. The prompts are taken from the "Mixture-of-Thoughts" dataset to ensure broad coverage of reasoning scenarios.

The SFT objective minimizes the negative log-likelihood of the teacher's trajectories under the student policy:

$$\mathcal{L}_{\text{SFT}}(\theta) = -\mathbb{E}_{(q, \boldsymbol{\tau}^*) \sim \mathcal{D}_{\text{teacher}}} \left[ \sum_{t=1}^{|\boldsymbol{\tau}^*|} \log \pi_\theta(y_t^* | q, \boldsymbol{y}_{<t}^*) \right] \tag{4}$$

During this stage, the vision encoder remains frozen to preserve visual capabilities while focusing alignment efforts on the language modeling components. This policy alignment stage stabilizes the student model and prepares it for the subsequent unified learning objective that combines reinforcement learning with on-policy distillation.

**Why Alignment Is Necessary.** A critical prerequisite for effective on-policy distillation is that the student and teacher models must have sufficiently overlapping output distributions. Without the initial alignment, several issues arise that inhibit the distillation process. The fundamental challenge stems from state-distribution shift: on-policy knowledge distillation evaluates KL divergence at prefixes $h_t \sim \pi_\theta$ sampled from the student's own policy. When $\pi_\theta$ is far from $\pi_\phi$'s support, the teacher's distributions at these off-distribution prefixes become diffuse and uninformative, yielding weak or high-variance gradients. Since reverse KL divergence is mode-seeking, it attempts to pull $\pi_\theta$ toward teacher modes at each $h_t$. However, when $h_t$ are off-distribution, the gradients may over-regularize irrelevant regions or destabilize training updates (see Sec.4.3 for a respective ablation).

The resulting distribution alignment ensures that student rollouts include states where the teacher has sufficient probability mass, making token-level KL divergence both informative for the training while providing stable gradients. From a formal perspective, minimizing $\mathbb{E}_{h_t \sim \pi_\theta}[D_{\text{KL}}(\pi_\phi \| \pi_\theta)]$ benefits significantly when $h_t$ corresponds to regions where $\pi_\phi$ has low entropy. The SFT phase increases the probability mass of such informative states under $\pi_\theta$, reducing gradient variance and improving optimization conditioning for the subsequent unified training stage.

**Stage 2: Unified RL and On-Policy Distillation**   Building on the aligned model from Stage 1, our core contribution is a unified objective that seamlessly combines reinforcement learning with teacher distillation. The key insight driving this approach is that both GRPO and on-policy knowledge distillation require sampling trajectories from the student policy $\pi_\theta$—the computationally expensive component of both training paradigms. By reusing the same rollouts for both objectives, we provide dense teacher guidance to the RL process with minimal computational overhead.

Our unified framework replaces the standard GRPO KL-divergence penalty against the old policy $\pi_{\theta_{\text{old}}}$ with a reverse KL-divergence term that pulls the student towards the teacher policy $\pi_\phi$. This substitution is motivated by recent findings (Liu et al., 2025b; Yu et al., 2025) that the reference policy KL regularization in GRPO can often be omitted without performance degradation, creating an opportunity to introduce teacher guidance at virtually no additional cost.

The unified objective combines both components into a single loss function:

$$\mathcal{L}_{\text{VOLD}}(\theta) = \mathcal{L}_{\text{GRPO}}(\theta) + \beta \cdot \mathbb{E}_{q, \boldsymbol{\tau} \sim \pi_\theta} \left[ \sum_{t=1}^{T} D_{\text{KL}} \left( \pi_\phi(\cdot|h_t) \| \pi_\theta(\cdot|h_t) \right) \right] \tag{5}$$

where $\beta > 0$ is the hyperparameter balancing reward maximization and teacher distillation. This framework naturally integrates exploration through RL with exploitation of teacher knowledge, operating on the same on-policy samples. The teacher provides token-level guidance on the student's own rollout prefixes, while the GRPO component drives the student towards high-reward solutions through trajectory-level binary rewards on verifiable text-only reasoning tasks. We extract final answers using structured prompting formats such as "boxed{...}" to compute the reward computation.

**Reward-Guided KL Masking**   A potential conflict arises between RL and distillation signals when the student discovers correct reasoning paths that diverge from the teacher's approach. To address this, inspired by (Xu et al., 2025b), we introduce reward-guided KL masking based on the principle of selective imitation by applying distillation only to incorrect responses, allowing the model to freely explore novel correct paths without teacher interference.

We implement this through response-level masking using the binary reward as a mask. Since our rewards are binary ($r(\boldsymbol{\tau}) \in \{0, 1\}$), the term $(1 - r(\boldsymbol{\tau}))$ naturally creates a mask that activates distillation only for failed attempts. This leads to our masked VOLD objective:

$$\mathcal{L}_{\text{VOLD -masked}}(\theta) = \mathcal{L}_{\text{GRPO}}(\theta) + \beta \cdot \mathbb{E}_{q, \boldsymbol{\tau} \sim \pi_\theta} \left[ (1 - r(\boldsymbol{\tau})) \sum_{t=1}^{T} D_{\text{KL}} \left( \pi_\phi(\cdot|h_t) \| \pi_\theta(\cdot|h_t) \right) \right] \tag{6}$$

When a rollout receives a positive reward ($r = 1$), the KL term is masked out (set to zero), allowing the student to retain its successful reasoning strategy. Conversely, teacher distillation remains active only for incorrect rollouts ($r = 0$), providing guidance when the student's approach fails.

## 4  EXPERIMENTS

### 4.1  EXPERIMENTAL SETUP

**Models and Checkpoints**   We use `Qwen2.5-VL-3B-Instruct` (3.75B parameters) as student VLM and `Qwen3-8B` as default text-only teacher LLM (see Appendix B.2 for more teacher sizes). Both models share the same tokenizer, satisfying the critical requirement for meaningful KL divergence computation during on-policy distillation. During fine-tuning we apply updates to the full parameter space of the language model while keeping the vision tower frozen throughout training.

**Training Data**   ◇For SFT, we use the Mixture-of-Thoughts (MoT)(Face, 2025) dataset, a curated collection of 350k verified reasoning traces spanning mathematics, coding, and science tasks. We take the prompts from MoT and generate new reasoning traces using our Qwen3-8B teacher model, creating the "MoT-Teacher-8B" dataset that captures the teacher's reasoning style and output distribution. We apply minimal postprocessing by keeping only trajectories under 8192 tokens for

Table 1: **Results on multimodal reasoning benchmarks:** VOLD achieves state-of-the-art performance despite training exclusively on text data, outperforming baselines that use images during fine-tuning. Baselines marked with $\ddagger$ were trained on portions of the evaluation set.

| Model | Images in FT | *Mutimodal General Tasks* | | *Multimodal Math* | | | | | *Visual IQ-Test* |
|---|---|---|---|---|---|---|---|---|---|
| | | MMMU-Pro (Vision) | MMStar | Math Vision | MathVista | MathVerse | DynaMath *(Avg)* | WeMath | LogicVista |
| Qwen2.5-VL-3B | - | 27.1 | 55.9 | 21.9 | 61.2 | 31.2 | 42.7 | 22.9 | 40.3 |
| XReasoner-3B *(repl.)* | ✗ | 31.0 | 55.2 | 24.4 | 61.1 | 35.7 | 47.2 | 30.6 | 41.1 |
| VLM-R1 3B-Math | ✓ | 28.6 | **56.7** | 21.9 | 62.7$^{\ddagger}$ | 32.2$^{\ddagger}$ | 42.7 | 30.0 | 40.5 |
| VLAA-Thinker 3B | ✓ | 24.6 | 55.6 | 24.4 | 61.0$^{\ddagger}$ | 36.4 | 47.5 | 31.5 | 38.5 |
| VOLD (Ours) | ✗ | **32.0** | 55.2 | **28.0** | 61.9 | **37.9** | **50.7** | **31.81** | **45.0** |

computational efficiency, without answer verification since the SFT stage aims purely for distributional alignment. ◇For RL, we use the text-only "orz-57k"(Hu et al., 2025) dataset containing mathematical problems with ground truth answers. We employ exact match verification to compute binary rewards by comparing generated answers against ground truth. For validation during RL training, we track progress on Geo3K(Lu et al., 2021), a visual geometry reasoning dataset, to monitor text-to-vision transfer from text-only training to visual reasoning tasks.

**Training & Implementation Details**  ◇For SFT, we train on the MoT-Teacher-8B corpus using batch size 256, learning rate $5 \times 10^{-5}$, and 4000 steps (approximately 5 epochs). ◇For RL, we apply GRPO on the text-only "orz-57k" dataset for 60 steps with KL coefficient $\beta = 0.1$, 5 rollouts per prompt, batch size 256, and learning rate $6 \times 10^{-6}$ (see details in Appendix).

**Evaluation Benchmarks**  We evaluate on a diverse suite of benchmarks categorized as follows: general multimodal reasoning benchmarks (MMMU-Pro(Wang et al., 2024c), MMStar(Chen et al., 2024)), multimodal math benchmarks (MathVista(Lu et al., 2023), MathVision(Wang et al., 2024b), MathVerse(Zhang et al., 2024), WeMath(Qiao et al., 2024), DynaMath(Zou et al., 2025)), and multimodal logic benchmarks (LogicVista(Xiao et al., 2024)). Detailed specifications including dataset versions, splits, and sample counts are provided in the supplementary material.

**Evaluation Details**  We use the vLLM(Kwon et al., 2023) as our inference engine with the widely adopted VLMEvalKit library(Duan et al., 2024) for standardized evaluation. Accuracy is reported for each dataset. Answer extraction employs GPT-4o-mini through the VLMEvalKit pipeline to parse model responses and extract final answers consistently across all evaluations.

## 4.2 RESULTS

**Baselines**  For fair comparison, we evaluate against baselines that *start from exactly the same base model*: Qwen2.5-VL-3B-Instruct. We compare against **X-Reasoner**, which is the only baseline using exclusively text-only training data—trained with similar datasets as our approach (the original MoT for SFT and orz-57k for RL) but without on-policy distillation. We carefully replicated X-Reasoner as no checkpoints were provided at the time of writing. We also evaluate against **VLAA-Thinker** and **VLM-R1-Math**, both trained using RL on image-text math datasets. Crucially, while VLAA-Thinker and VLM-R1-Math use images during fine-tuning and RL, our approach trains exclusively on text-only data, enabling broader scalability through text-rich reasoning resources.

**SOTA Comparison**  The results in Table 1 demonstrate that VOLD achieves substantial performance improvements across most benchmarks, with the most significant gains observed on challenging reasoning tasks. On MathVision, VOLD achieves 28.0%, outperforming VLAA-Thinker (24.4%) and the base model (21.9%) by substantial margins. Similarly, on LogicVista, VOLD reaches 45.0% compared to VLM-R1's 40.5% and the base model's 40.3%.

The comparison with X-Reasoner, which uses the same text-only training approach with identical datasets (original MoT for SFT and orz-57k for RL), reveals the critical importance of on-policy distillation. While X-Reasoner achieves modest improvements over the base model, VOLD 's unified RL+distillation approach yields substantial gains across all benchmarks (e.g., 32.0% vs 31.0% on

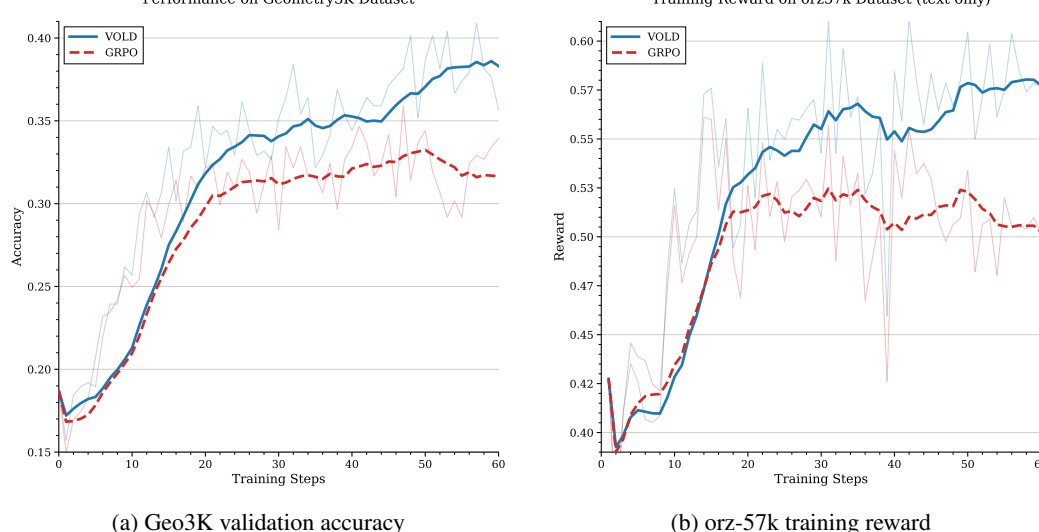

(a) Geo3K validation accuracy           (b) orz-57k training reward

Figure 3: **Learning dynamics:** *(left)*: Accuracy on the visual Geo3K dataset. *(right)*: Reward on the text-only orz-57k training data. The results show a significant gain by using VOLD.

Table 2: **Policy Alignment Ablation:** demonstrates the critical role of aligning the student with the teacher's output distribution. We compare our full method, which uses teacher-generated SFT data for alignment, against variants trained on the original MoT dataset, creating a policy mismatch. The results show that without proper alignment, on-policy distillation provides no additional benefit.

| Components | | | Dataset Performance | | | | | | | |
|---|---|---|---|---|---|---|---|---|---|---|
| SFT MoT | RL | On-Policy Dist. | MMMU-Pro | MMStar | Mathvision | MathVista | MathVerse | DynaMath$_{(Avg.)}$ | WeMath | LogicVista |
| ✗ | ✗ | ✗ | 27.1 | 55.9 | 21.9 | 61.2 | 31.2 | 42.7 | 22.9 | 40.3 |
| ✓ | ✗ | ✗ | 27.3 | 54.1 | 22.0 | 59.1 | 31.3 | 42.4 | 21.4 | 38.0 |
| ✗ | ✓ | ✗ | 27.5 | 55.2 | 23.8 | 61.2 | 31.2 | 46.7 | 24.6 | 40.1 |
| ✓ | ✓ | ✗ | 31.0 | 55.2 | 24.4 | 61.1 | 35.7 | 47.2 | 30.6 | 41.1 |
| ✓ | ✓ | ✓ | 30.8 | 55.1 | 24.5 | 61.0 | 35.9 | 47.4 | 30.6 | 41.2 |
| VOLD (ours) | | | **32.0** | **55.2** | **28.0** | **61.9** | **37.9** | **50.7** | **31.8** | **45.0** |

MMMU-Pro, 28.0% vs 24.4% on MathVision). This validates our hypothesis that teacher guidance during RL exploration is essential for efficient text-to-vision knowledge transfer.

Remarkably, VOLD outperforms methods that train directly on image-text data despite using exclusively text-only training. However, fair comparison is complicated by dataset contamination issues inherent to visual reasoning training. VLM-R1-Math was specifically trained on MathVista, explaining its strong MathVista performance (62.7%). VLAA-Thinker was trained on filtered collections from several benchmarks, with approximately 40% of images overlapping with evaluation sets due to the scarcity of high-quality visual reasoning data. Despite these advantages for image-trained baselines, VOLD demonstrates the effectiveness of leveraging rich text-based reasoning resources for text-to-vision transfer.

**Learning Dynamics** Figure 3 compares VOLD against vanilla GRPO during RL training, tracking both training reward on the text-only orz-57k dataset and validation accuracy on the visual Geo3K dataset. Both methods start from the same SFT checkpoint. VOLD consistently outperforms GRPO on both metrics: achieving 0.58 vs 0.51 training reward and 0.38 vs 0.32 Geo3K accuracy. Notably, the simultaneous improvement on visual reasoning tasks despite text-only training demonstrates successful text-to-vision knowledge transfer. The widening performance gap over training steps indicates that on-policy distillation provides increasingly valuable guidance, leading to more stable and effective reasoning transfer compared to pure the vanilla GRPO approaches.

## 4.3 ABLATIONS

**Stage 1/Policy Alignment Ablation** To demonstrate the importance of Stage 1 policy alignment in our framework, we compare different training configurations where the student model is aligned

Table 3: **Component Analysis of VOLD :** This table isolates the contribution of each component in our two-stage framework. We show performance after SFT-only, after adding RL (GRPO), and with our full unified objective. While Stage 1 SFT aligns the policy, it temporarily degrades performance due to unfiltered teacher traces. Stage 2, which combines RL with on-policy distillation, provides the largest performance gains, demonstrating that both components are essential for optimal reasoning.

| Components | | | Dataset Performance | | | | | | | |
|---|---|---|---|---|---|---|---|---|---|---|
| SFT Teacher-MoT | RL | On-Policy Dist. | MMMU-Pro | MMStar | Mathvision | MathVista | MathVerse | DynaMath$_{(Avg.)}$ | WeMath | LogicVista |
| ✓ | ✗ | ✗ | 25.8 | 49.7 | 18.6 | 55.1 | 27.8 | 42.1 | 30.4 | 28.9 |
| ✓ | ✓ | ✗ | 29.65 | 50.47 | 24.01 | 58.4 | 34.14 | 47.62 | 21.43 | 38.255 |
| ✓ | ✓ | ✓ | **31.96** | **55.2** | **28.0** | **61.9** | **38.0** | **50.7** | **31.8** | **45.0** |

with different reasoning trace sources. Instead of using MoT-Teacher-8B (generated by our Qwen3-8B teacher), we train on the original Mixture-of-Thoughts dataset where reasoning traces were generated by DeepSeek-R1. As shown in Table 2, training directly on the original MoT dataset does not degrade performance as severely as training on MoT-Teacher-8B, since the original dataset was filtered to retain only traces leading to correct answers and its composition was optimized for downstream performance. However, despite this stronger starting point, the pipeline fails to benefit from on-policy distillation during Stage 2, achieving nearly identical performance whether using SFT+RL, RL-only or SFT+RL+Distillation. This lack of improvement occurs because the student model remains misaligned with the teacher's distribution, preventing effective knowledge transfer. In contrast, our complete VOLD pipeline with proper teacher-student alignment achieves substantial gains, confirming that Stage 1 policy alignment is essential for enabling effective on-policy distillation in Stage 2.

**Component Analysis of VOLD** To isolate the contribution of each component in our framework, we compare the full VOLD pipeline against its constituent parts across diverse benchmarks. Table 3 presents results for three configurations: SFT-only training on MoT-Teacher-8B, SFT followed by RL-only (without on-policy distillation), and our complete VOLD. The full VOLD pipeline consistently achieves the best performance across all benchmarks, with particularly notable improvements on MMStar (55.2% vs 50.47%), MathVision (27.96% vs 24.01%), and LogicVista (44.96% vs 38.26%). Interestingly, the SFT-only model shows degraded performance compared to the base model, which we attribute to the inclusion of unfiltered teacher reasoning traces that may contain incorrect reasoning paths leading to wrong answers. Despite this initial performance drop, the SFT phase remains essential for establishing distributional alignment that enables effective teacher guidance during subsequent RL training with on-policy distillation. The RL-only configuration achieves moderate improvements over SFT-only but falls short of VOLD's performance, confirming that neither component alone is sufficient for optimal reasoning transfer. We leave the exploration of more sophisticated SFT dataset filtering strategies for future work, as removing incorrect traces would significantly increase the computational cost of generating the initial training corpus.

In the Appendix, we provide additional ablations analyzing the impact of SFT durationB and teacher model sizeB.2, finding that sufficient initial alignment is a crucial prerequisite and that teacher performance gains show diminishing returns.

## 5 CONCLUSION

We introduced VOLD, a unified framework for transferring reasoning capabilities from text-only teacher models to vision-language students through combined RL and On-Policy Distillation. Our approach addresses the fundamental challenge of training VLMs for reasoning tasks without requiring scarce and expensive visual reasoning data. The key insight driving our framework is that effective text-to-vision reasoning transfer requires initial policy alignment between teacher and student models. Our two-stage pipeline first establishes this alignment through SFT on teacher-generated reasoning traces, then applies unified RL and distillation objectives to enhance reasoning capabilities while maintaining teacher guidance throughout exploration. VOLD represents a significant step toward scalable VLM reasoning training by leveraging abundant text-based reasoning resources rather than relying on limited visual reasoning data. The framework is orthogonal to advances in RL algorithms and can seamlessly integrate with improved methods beyond GRPO, offering a promising direction for future research in text-to-vision reasoning transfer and multimodal reasoning.

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

# APPENDIX

## A  IMPLEMENTATION DETAILS

Detailed training hyperparameters for both stages of the VOLD  framework are provided in Table 4. SFT experiments were conducted using 32 A100 GPUs and RL on 4 A100 GPUs, with gradient accumulation to achieve the specified effective batch sizes.

Table 4: Training hyperparameters for Stage 1 (SFT) and Stage 2 (RL) in the VOLD  framework.

| Hyperparameter | Stage 1 (SFT) | Stage 2 (RL) |
|---|---|---|
| Learning Rate | $5 \times 10^{-5}$ | $6 \times 10^{-6}$ |
| Batch Size | 256 | 256 |
| Training Steps | 4000 | 60 |
| Rollouts per Prompt | - | 5 |
| KL Coefficient ($\beta$) | - | $1 \times 10^{-3}$ |
| Clipping Threshold ($\epsilon_{\text{upper}}$) | - | 0.3 |
| Clipping Threshold ($\epsilon_{\text{lower}}$) | - | 0.2 |
| Max Sequence Length | 8192 | 8192 |
| Optimizer | AdamW | AdamW |
| Weight Decay | $1 \times 10^{-2}$ | $1 \times 10^{-2}$ |
| Warmup Steps | 150 | 5 |
| LR Schedule | Cosine | Cosine |
| Vision Encoder | Frozen | Frozen |

## B  MORE ABLATIONS

### B.1  IMPACT OF COLD START

To understand when on-policy distillation becomes beneficial and how SFT duration affects subsequent RL training effectiveness, we evaluate different cold start checkpoints from various stages of teacher-trace SFT on MoT-Teacher-8B. We apply identical RL training with on-policy distillation to checkpoints saved at SFT steps 500, 1000, 1500, 2000, 2500, 3000, 3500, and 4000. Figure 4 shows the resulting training dynamics for both validation accuracy on Geo3K (left) and training reward (right), where the color gradient represents different starting checkpoints: light red corresponds to 500 SFT steps, progressing through darker reds to black representing 4000 SFT steps.

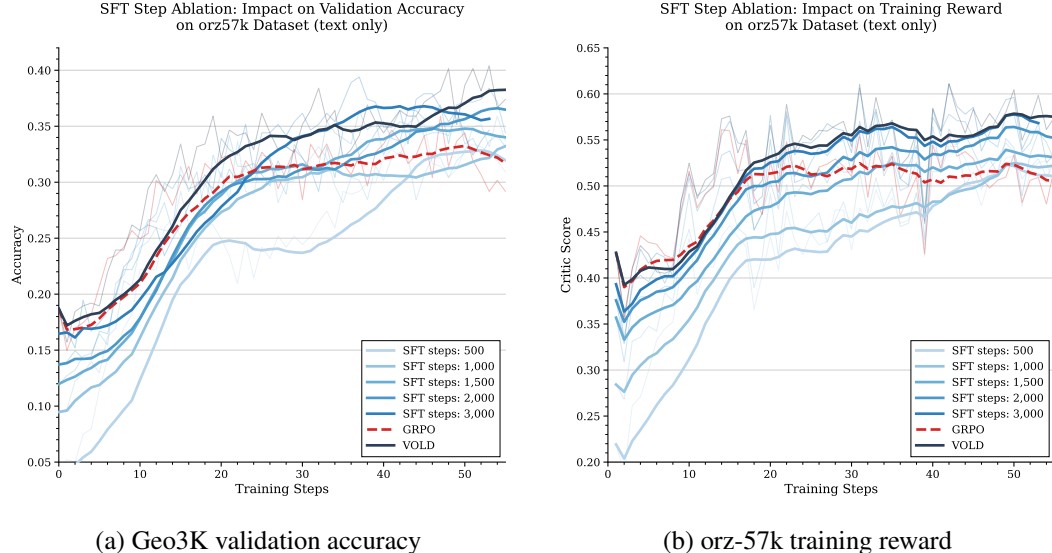

(a) Geo3K validation accuracy     (b) orz-57k training reward

Figure 4: **Sufficient Policy Alignment is Crucial for On-Policy Distillation**. This figure illustrates that the benefit of our unified objective depends on the quality of the initial alignment from Stage 1. Models with short SFT phases (light blue) are poorly aligned with the teacher and fail to benefit from its guidance. As the alignment improves with more SFT steps (darker blue), the student can better leverage the on-policy distillation signal, unlocking significant performance gains over the GRPO-only baseline (red).

The yellow curve shows the GRPO-only baseline starting from the 4000-step checkpoint. Early SFT checkpoints (light red, e.g., step 500) initially perform worse than the base model and show no benefit from distillation, indicating that minimal teacher-trace exposure is insufficient for distributional alignment. Performance progressively improves as SFT training continues, with the student's output distribution gradually converging toward the teacher's, effectively establishing a "breadcrumb trail" that guides subsequent RL exploration. Later checkpoints (darker colors approaching black) demonstrate substantial improvements over the GRPO-only baseline, as the student can better follow the teacher's reasoning patterns through on-policy distillation. The dynamics plateau after approximately 3000 SFT steps, suggesting that the student has sufficiently internalized the teacher's reasoning style and established a robust breadcrumb trail for RL guidance. These results demonstrate that adequate cold start training is crucial for creating the distributional bridge necessary for successful knowledge transfer during the unified RL+On-policy Distillation phase.

Table 5: **Impact of Teacher Model Size**. We evaluate the final performance of VOLD when using teacher models of varying scales (4B, 8B, and 14B parameters). While increasing teacher size from 4B to 8B yields performance gains across most benchmarks, we observe diminishing returns with the 14B teacher, which provides no consistent improvement over the 8B model.

| Dataset | Teacher 4B | Teacher 8B | Teacher 14B |
|---|---|---|---|
| MMMU-Pro | 32.60 | 31.96 | 32.21 |
| MMStar | 53.66 | 55.2 | 55.1 |
| Mathvision | 26.14 | 27.96 | 27.81 |
| MathVista | 61.5 | 61.9 | 62.0 |
| MathVerse | 39.21 | 37.94 | 38.31 |
| DynaMath_(Average) | 48.95 | 50.66 | 50.93 |
| WeMath | 30.29 | 31.81 | 31.58 |
| LogicVista | 42.95 | 44.97 | 44.85 |

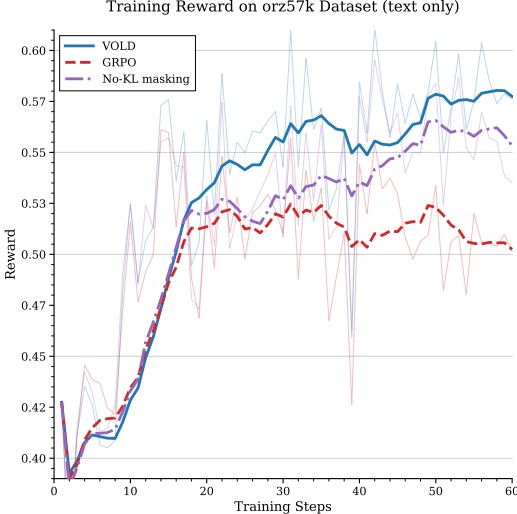

Figure 5: Training reward comparison: VOLD with KL masking (blue), without masking (purple), and vanilla GRPO (red). KL masking provides consistent performance gains throughout training.

## B.2 TEACHER SIZE ABLATION

To investigate the impact of teacher capacity on student performance, we compare VOLD training using different teacher model sizes during the RL+KD phase. We evaluate three teacher configurations: Qwen3-4B, Qwen3-8B, and Qwen3-14B, all sharing the same tokenizer with the Qwen2.5-VL-3B student to enable meaningful KL divergence computation. Table 5 presents results across representative benchmarks for each teacher size. The results demonstrate that larger teacher models generally provide better guidance, with the 8B teacher outperforming the 4B teacher on most benchmarks, particularly on MMStar (55.2% vs 53.66%) and LogicVista (44.97% vs 42.95%). This improvement can be attributed to the superior reasoning capabilities of larger teachers, which provide more valuable supervision during on-policy distillation. However, the performance gains begin to saturate beyond the 8B scale, with the 14B teacher showing similar performance compared to the 8B variant on certain tasks. This saturation suggests that the 3B student model has inherent capacity limitations that prevent it from fully leveraging the guidance from very large teachers. The student's reasoning capabilities appear to plateau once the teacher reaches sufficient quality, indicating there is only so much knowledge the student can effectively absorb through on-policy distillation given its architectural constraints.

## B.3 EFFECT OF REWARD-GUIDED KL MASKING

To evaluate the impact of our reward-guided KL masking mechanism, we compare three configurations during RL training: vanilla GRPO, VOLD with KL masking (our full method), and VOLD without KL masking. As shown in Figure 5, the reward-guided masking provides a clear benefit over both alternatives. While VOLD without masking (purple line) outperforms vanilla GRPO (red line), achieving approximately 0.56 vs 0.51 final reward, our complete VOLD framework with KL masking (blue line) achieves the highest performance at 0.58. The consistent gap throughout training demonstrates that selectively applying distillation only to incorrect responses allows the model to retain successful reasoning strategies while still benefiting from teacher guidance on failed attempts. This validates our hypothesis that masking prevents interference between RL exploration of novel correct paths and teacher distillation objectives.

