# OpenReview forum: "VOLD: Reasoning Transfer from LLMs to Vision-Language Models via On-Policy Distillation"
_ICLR.cc/2026/Conference — ICLR 2026 Conference Withdrawn Submission_

### Official Review · Reviewer_hzCZ · 2025-10-27

**Soundness:** 2
**Presentation:** 3
**Contribution:** 2
**Rating:** 4
**Confidence:** 4

**Summary:**

The paper proposes VOLD, a two-stage framework for transferring reasoning abilities from text-only LLMs to VLMs without using visual reasoning data. In Stage 1, the VLM is aligned with the teacher LLM through supervised fine-tuning on teacher-generated reasoning traces to ensure distributional consistency. In Stage 2, a unified objective combines GRPO with on-policy knowledge distillation, where the teacher guides the student using reverse KL divergence on the student’s own rollouts. A reward-guided KL masking mechanism applies distillation only to incorrect responses. Experiments on multimodal benchmarks show that VOLD achieves excellent performance among text-only trained methods and rival approaches trained with visual data. Ablations confirm the necessity of initial policy alignment and the benefit of the unified RL–distillation objective.

**Strengths:**

1. The method is well-formulated, with clear mathematical definitions and a coherent training pipeline. Ablation studies and component analyses validate the necessity of each design choice.

2. The paper is clearly structured and easy to follow, explaining both theoretical motivations and empirical findings with sufficient detail

**Weaknesses:**

1. The related work section omits several near-contemporary methods in teacher-guided or reasoning-enhanced multimodal training, which weakens the positioning of its originality claim. Specifically, it does not sufficiently distinguish itself from recent concurrent works that explore similar ideas of reasoning transfer or guided RL training, such as:

[1] Yan, J., Li, Y., Hu, Z., Wang, Z., Cui, G., Qu, X., Cheng, Y., & Zhang, Y. (2025, April 21). Learning to Reason under Off-Policy Guidance. arXiv preprint arXiv:2504.14945

[2] Shen, J., Zhao, H., Gu, Y., Gao, S., Liu, K., Huang, H., Gao, J., Lin, D., Zhang, W., & Chen, K. (2025, July 22). Semi-off-Policy Reinforcement Learning for Vision-Language Slow-Thinking Reasoning. arXiv preprint arXiv:2507.16814

The motivation and design are similar to those in this paper. While direct empirical comparison may not be feasible, a deeper conceptual discussion on how VOLD differs (e.g., the implications of on-policy vs semi-off-policy guidance, or different stability mechanisms) is missing.

2. The experiments rely on a single base model. Evaluating across additional backbones or model families (e.g., InternVL, LLaVA or Phi-VL) would increase robustness and show the generality of the framework.

3. A more systematic exploration of how teacher capacity, reasoning quality, and policy similarity affect transfer efficiency would add depth and interpretability to the experimental results.

**Questions:**

See above weaknesses.

---

### Official Review · Reviewer_rbRb · 2025-10-30

**Soundness:** 2
**Presentation:** 3
**Contribution:** 2
**Rating:** 4
**Confidence:** 4

**Summary:**

This paper addresses the challenge of training vision-language models (VLMs) for complex reasoning tasks, which is often hindered by the scarcity of high-quality, image-text reasoning data. The authors propose VOLD, a novel two-stage framework to transfer reasoning capabilities from a text-only large language model (LLM) teacher to a VLM student.

The core contributions are:

1. A Two-Stage Training Pipeline:

    **Stage 1:** Policy Alignment. The VLM student is first fine-tuned via Supervised Fine-Tuning (SFT) on a large corpus of reasoning traces generated by the text-only teacher. The paper argues this "cold-start" alignment is critical to bridge the distributional gap between the teacher and student, enabling effective subsequent training.

    **Stage 2:** Unified On-Policy Training. The aligned student is then trained using a unified objective that combines reinforcement learning (specifically, Group Relative Policy Optimization or GRPO) with on-policy knowledge distillation. This allows the student's reasoning process to be guided by the teacher's feedback on its own generated trajectories (on-policy), improving sample efficiency and reasoning quality. The unified loss is formulated as:$L_{\text{VOLD}} (\theta) = L_{\text{GRPO}}(\theta) + \beta \cdot \mathbb{E}{q,\tau \sim \pi\theta} \left[ \sum_{t=1}^{T} D_{\text{KL}} (\pi_\phi(\cdot|h_t) | \pi_\theta(\cdot|h_t)) \right]$

2. Reward-Guided KL Masking: An enhancement where the distillation loss is only applied to trajectories that result in an incorrect answer (reward=0). This prevents the teacher from penalizing novel, correct reasoning paths discovered by the student.

3. Empirical Validation: The authors conduct extensive experiments on multiple challenging multimodal reasoning benchmarks (e.g., MMMU-Pro, MathVision, LogicVista). The results show that VOLD significantly outperforms the base VLM and a strong baseline (X-Reasoner) that uses RL without distillation. Notably, VOLD achieves state-of-the-art performance compared to several models that were fine-tuned on in-domain image-text data, despite VOLD being trained exclusively on text-only data. Thorough ablation studies validate the necessity of the policy alignment stage and the effectiveness of each component of the VOLD framework.

**Strengths:**

1. Novel and Effective Framework: The core strength is the VOLD framework itself—a well-motivated and coherent two-stage process that effectively combines SFT, RL, and knowledge distillation for cross-modal reasoning transfer.

2. Thorough and Convincing Ablation Studies: The paper's quality is significantly elevated by its rigorous ablation studies. The experiments systematically validate the necessity of the policy alignment stage (Table 2), the contribution of each component (Table 3), the impact of SFT duration (Fig. 4), and the effect of teacher size (Table 5). This level of detailed analysis provides strong support for the authors' claims and offers deep insights into the method's mechanics.

3. Strong Empirical Results: VOLD achieves state-of-the-art results on several challenging benchmarks, outperforming not only text-trained baselines but also models trained directly on in-domain visual data. This demonstrates the high practical efficacy of the proposed approach.

4. High Scalability: By relying exclusively on text-only data for reasoning training, the proposed method is highly scalable and circumvents the major bottleneck of curating large-scale visual reasoning datasets. This makes the approach very practical and impactful.

**Weaknesses:**

1. Generalizability to Heterogeneous Models: The method's reliance on a shared tokenizer between the teacher and student is a key requirement for the KL divergence calculation. This is explicitly mentioned but also means the study is confined to the Qwen model family. This raises questions about the framework's generalizability to scenarios where one might want to use a teacher and student from different model families (e.g., a GPT-4 teacher and a Llama-based VLM student). A discussion of potential solutions or the challenges involved (e.g., vocabulary projection) would strengthen the paper.

2. Nuance of the SFT Stage: The paper interestingly notes that SFT on unfiltered teacher traces initially degrades performance compared to the base model (Table 3). The authors attribute this to incorrect reasoning paths in the teacher's outputs. This is an important finding but could be explored further. It's unclear what the error rate of the teacher is and whether the SFT stage is primarily learning reasoning style over correctness. A brief analysis of the teacher's trace quality would provide valuable context.

3. Computational Overhead: The on-policy distillation stage requires querying the teacher model at each step of the student's rollout. Given that the teacher is a larger model (e.g., 8B), this seems computationally expensive compared to a standard RL fine-tuning that only uses a reward function and a reference model. A brief discussion or analysis of the computational overhead of VOLD compared to the baselines would be beneficial for practitioners.

**Questions:**

1. The requirement of a shared tokenizer is a practical limitation. How do the authors envision adapting VOLD to a setting with a heterogeneous teacher-student pair (e.g., different model families)? What specific challenges would arise for the on-policy $D_{\text{KL}}$ computation, and are there promising directions to overcome them?

2. The finding that SFT on unfiltered teacher traces can degrade performance is very insightful. Could you elaborate on the trade-off here? Does this suggest that the primary role of Stage 1 is to align the student's output format and reasoning style with the teacher, even at the cost of initial accuracy, to create a better substrate for the Stage 2 distillation? Have you analyzed the error rate in the "MoT-Teacher-8B" dataset?

3. The reward-guided KL masking is an effective binary mechanism. Did you consider or experiment with any "soft" masking approaches? For instance, could the KL loss be weighted by a continuous value, such as the advantage calculated by GRPO or another confidence metric, to provide a more graded application of teacher guidance?

---

### Official Review · Reviewer_rFkZ · 2025-10-30

**Soundness:** 2
**Presentation:** 4
**Contribution:** 3
**Rating:** 4
**Confidence:** 3

**Summary:**

This paper presents a framework called VOLD for transferring reasoning abilities from text-only models to VLM student models. VOLD contains a warm up phase to align doctor and teacher policy with SFT, followed by the second training stage that combines reinforcement learning with on-policy distillation. Experiments show that VOLD has shown remarkable performance across diverse benchmarks including MMMUPro, MathVision, MathVista, and LogicVista.

**Strengths:**

1. The paper is well written.
2. The motivation is clear: on-policy distillation enables effective transfer of reasoning capabilities from text-only domain to vision language task. The SFT "cold-start" aligns the distribution between teacher and student.
3. The proposed method is simple yet remains effective according to the empirical results
4. Comprehensive ablations are presented to analyze the proposed method

**Weaknesses:**

1. **Limited novelty**. Distillation from large language models has been widely studied on various tasks (including Qwen3) [1,2,3]. VOLD presents a simple combination of normal RL training and on-policy distillation. Besides, recent work [4] also studies the transfer between text to visual reasoning with a two-stage recipe (SFT for stage one and RL for stage two). These works raise my concern about the novelty and contribution of the method.
2. **Lack of analysis** about the behavior of finetuned model. The evaluation in the paper is heavily focused on the numerical results (and mostly on math data). In-depth analysis is missing: for multimodal reasoning, what can be learned and what cannot be learned from VOLD? What "reasoning transfer" actually covers for visual reasoning tasks?
3. **Tokenizer constraint**. As described by the paper, the teacher and student must share the same tokenizer, which limits the applicability of VOLD.

[1] Qwen3 Technical Report. arXiv: 2505.09388

[2] On-policy Distillation of Language Models: Learning from Self-generated Mistakes. ICLR 2024.

[3] OpenThoughts: Data Recipes for Reasoning Models. arXiv: 2506.04178

[4] Open Vision Reasoner: Transferring Linguistic Cognitive Behavior for Visual Reasoning. NeurIPS, 2025

**Questions:**

1. What's the range of visual reasoning tasks that VOLD could cover? For example, for visual reasoning tasks that heavily relies on perception, would the model trained from VOLD behave well or not?

2. A recent work [4] presents a two-stage method to transfer the reasoning ability, with SFT on stage 1 and RL on stage 2. What's the conceptual and performance difference between [4] and VOLD?

3. What's the performance of only doing on-policy distillation, at Table 2 and 3? Recent works show that on-policy distillation alone would yield strong performance [5]

4. What's the exact value of beta used in the paper? In Section 4.1 line 344 it's said $\beta=0.1$, while in Table 4 it's reported that $\beta=1e-3$

[4] Open Vision Reasoner: Transferring Linguistic Cognitive Behavior for Visual Reasoning. NeurIPS, 2025

[5] On-Policy Distillation. Thinking Machines Lab: Connectionism, Oct 2025.

---

### Official Review · Reviewer_hTxZ · 2025-10-31

**Soundness:** 1
**Presentation:** 2
**Contribution:** 1
**Rating:** 2
**Confidence:** 5

**Summary:**

This paper proposes VOLD, a two-stage framework to improve the reasoning capabilities of Vision-Language Models (VLMs) by training them exclusively on text-only reasoning data.

**Strengths:**

1. the paper correctly identifies policy alignment as a premise for on-policy distillation

**Weaknesses:**

1. The paper studies whether complex multimodal reasoning can be significantly improved by training only on textual reasoning data. The authors motivate this by the scarcity of multimodal data, but they fail to address the necessity of it.

- From both an intuitive and logical perspective, multimodal reasoning is not just textual reasoning with an image attached; it is an integrated task where visual perception and abstract reasoning are inextricably linked.

- In human cognition, perception informs reasoning; they should be both necessary. The paper's premise, however, treats textual reasoning (lacking perception) as a sufficient substitute for this integrated process. It fails to provide a convincing conceptual argument for why a text-only teacher, which is blind to the perceptual information, can effectively guide a multimodal model on tasks where visual perception is critical.

The paper does not justify why it is worthwhile using only text dataset, given that there are already tons of new high-quality multimodal reasoning dataset as proposed in MM-Eureka, VL-Rethinker, R1-Onevision, VLAA-Thinking, SRPO, etc.

2. Techinical novelty limited, and the claim of "Reasoning Transfer" does not align with the essence of the proposed technical mechanism.
- On-policy distillation is a well-established approach. When applying the approach to transfer textual reasoning to multimodal reasoning, there should be unique challenges from the visual side, but the authors do not explore it, leaving the technical contribution limited.
- The paper's technical mechanism (Eq. 6) actively masks out the distillation loss on all correct trajectories. This means the student is not being trained to imitate the teacher's correct reasoning style. It essentially receives token rewards on incorrect trajectories using the text-only teacher's token log-prob. Given this core functionality, the claim of textual reasoning transfer is weaker than presented.

3. Unsupported "Complementary" Claim: The authors claim the method is "orthogonal and complementary" to other approaches. This is a significant claim that is left entirely unsupported by experimental evidence. No experiments are run to show that VOLD have complementary benefits when training with sufficient multimodal reasoning dataset.

4. The paper do not convincingly show the significance of textual reasoning transfer without comparing with strong SOTA VLM RL approaches, such as MM-Eureka, R1-Onevision, VL-Rethinker, SRPO, etc.

5. Return-over-Investment: The proposed approach requires SFT training to align teacher-student policy distribution, and serving the teacher model during training. Given this extensive computation overhead, it remains unclear how the benefits of textual reasoning transfer outweigh that of direct RL training with multimodal data.

Given the clear flaws of the current manuscript, I do not recommend for acceptance.

**Questions:**

On Motivation: Given that multimodal reasoning intrinsically requires grounding in visual perception, could you provide a stronger conceptual justification for why a text-only teacher is a suitable guide, rather than simply a source of scalable (but modality-mismatched) data? Why should we expect this to work on anything beyond tasks where the visual component is trivial?

On Method (Eq. 6): Since the distillation loss is masked for all correct trajectories, how can the method claim to "transfer the teacher's reasoning ability"? Isn't it more accurate to describe this as a fine-grained reward signal for incorrect trajectories, which doesn't actually teach the student how the teacher reasons correctly?

On Experimental Design: Why was joint training (mixing VOLD's text-only data with multimodal reasoning data) not explored? This seems like a more logical approach to leverage both data sources and would be the most direct test of your "complementary" claim.

On Baselines: How does VOLD compare against a simpler baseline, such as standard SFT on the text-only data (Stage 1) followed by GRPO on multimodal reasoning data (like the data used by VLAA-Thinker or VLM-R1)? This would help isolate whether the text-only RL stage is truly beneficial over a multimodal RL stage.

---

### Official Review · Reviewer_kYLy · 2025-11-01

**Soundness:** 2
**Presentation:** 3
**Contribution:** 2
**Rating:** 4
**Confidence:** 3

**Summary:**

The paper proposes VOLD, a two-stage framework to transfer reasoning skills from a text-only teacher LLM to a vision-language student model (VLM) using only text data.
Stage 1 performs cold-start policy alignment via SFT on teacher-generated reasoning traces to align the student distribution with the teacher.
Stage 2 conducts unified on-policy training that combines GRPO-based RL with on-policy distillation (reverse KL) from the teacher, with reward-guided KL masking to avoid penalizing correct but teacher-divergent solutions.
Empirically, VOLD uses Qwen2.5-VL-3B as the student and Qwen3-8B as the teacher, and is evaluated zero-shot on multiple multimodal reasoning benchmarks. VOLD outperforms both the base model and text-only transfer baselines, and is competitive or superior to methods trained with image-text reasoning data, despite using only text in training.
Ablations demonstrate (i) the necessity of the SFT cold start for successful on-policy distillation, (ii) consistent gains from the unified RL+KD objective over RL alone, (iii) diminishing returns with larger teachers beyond 8B, and (iv) benefits of reward-guided KL masking.

**Strengths:**

Clear, principled framework: Unifies RL (GRPO) with on-policy distillation on shared rollouts, offering dense token-level guidance at minimal extra cost.

Strong empirical results: Consistent improvements across diverse multimodal benchmarks, including MathVision and LogicVista, despite training exclusively on text.

Careful ablations:
Cold-start alignment analysis convincingly shows that distributional alignment is a prerequisite for effective on-policy distillation.
Component analysis isolates the contributions of SFT, RL, and KD.
Teacher-size study and KL masking study provide practical insights.

Orthogonality: Method can integrate with improved RL algorithms beyond vanilla GRPO.

**Weaknesses:**

Limitations of visual training and performance ceiling: The proposed method primarily targets the transfer and enhancement of reasoning patterns, with training signals derived entirely from the text domain (the RL stage also uses purely text-based, verifiable tasks). It does not explicitly strengthen visual representations or cross-modal alignment. Consequently, for perception-intensive tasks that rely on fine-grained visual perception and spatial relationship modeling, the performance ceiling and generalization ability remain uncertain. In other words, gains along the “reasoning” dimension may not fully compensate for potential shortcomings along the “perception” dimension.

Paradigm choice for text-only reasoning enhancement: Empirically, improving a multimodal model’s text-based reasoning can yield substantial gains on benchmarks where reasoning dominates. However, this strategy raises a paradigm-level question: if the ultimate benefits primarily stem from text-side reasoning enhancement, is it more efficient and robust to build the VLM upon a stronger text backbone from the outset, rather than retrofitting it later via text-only RL and distillation?

Attribution fairness under teacher involvement: The explicit introduction of a teacher model in training raises two attribution challenges: (i) to what extent are the observed gains attributable to the teacher’s capability versus the proposed unified objective (GRPO + on-policy distillation) and training mechanism; and (ii) the current comparisons against baselines may be insufficient to disentangle the relative contributions of “teacher strength/quality” and “algorithmic design.” Finer-grained ablations and comparisons against other strong teacher-based baselines would strengthen the causal persuasiveness of the conclusions.

Completeness of related work coverage: Recent studies have also explored using text-only teachers as online mentors to improve VLM reasoning—for example, prompting the VLM to first produce structured/semantic image descriptions, followed by reasoning and guidance from a text LLM, thereby enabling text-teacher distillation [1]. Including and systematically contrasting such approaches would better position this work’s contributions and differences, and enhance the completeness of the research context.

[1] Semi-off-Policy Reinforcement Learning for Vision-Language Slow-Thinking Reasoning

**Questions:**

See Weakness.

---

### Note · Authors · 2025-11-14

**Comment:**

We thank all reviewers for their helpful comments and will consider them for a revised version.

**Withdrawal Confirmation:**

I have read and agree with the venue's withdrawal policy on behalf of myself and my co-authors.